# Scaled Inverse Graphics: Efficiently Learning Large Sets of 3D Scenes

## Abstract

While the field of inverse graphics has been witnessing continuous growth, techniques devised thus far predominantly focus on learning individual scene representations. In contrast, learning large sets of scenes has been a considerable bottleneck in NeRF developments, as repeatedly applying inverse graphics on a sequence of scenes, though essential for various applications, remains largely prohibitive in terms of resource costs. We introduce a framework termed "*scaled* inverse graphics", aimed at efficiently learning large sets of scene representations, and propose a novel method to this end. It operates in two stages: (i) training a compression model on a subset of scenes, then (ii) training NeRF models on the resulting smaller representations, thereby reducing the optimization space per new scene. In practice, we compact the representation of scenes by learning NeRFs in a latent space to reduce the image resolution, and sharing information across scenes to reduce NeRF representation complexity. We experimentally show that our method presents both the lowest training time and memory footprint in scaled inverse graphics compared to other methods applied independently on each scene. Our codebase is publicly available as open-source.

## 1 Introduction

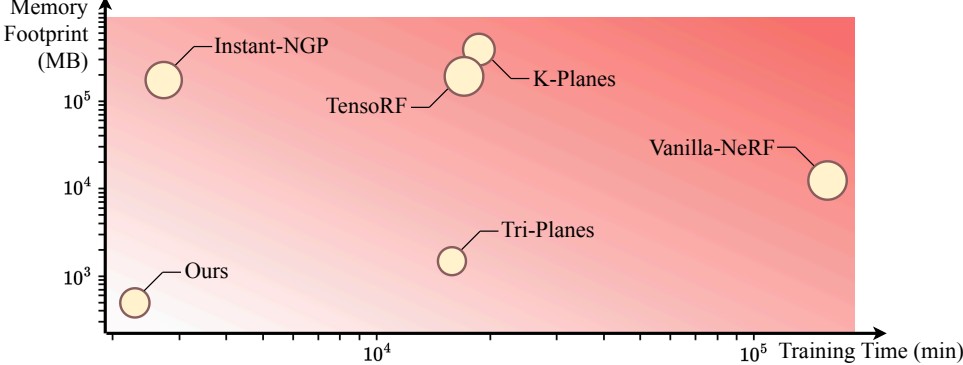

Figure 1: **Resource Costs.** Comparison of resource costs and novel view synthesis (NVS) quality of recent works when naively scaling the inverse graphics problem ($N = 2000$ scenes). Circle sizes represent the NVS quality of each method. Our method presents similar NVS rendering quality compared to Tri-Planes, our base representation, while demonstrating the lowest training time and memory footprint of all methods. The data behind this figure can be found in Appendix A.

The inverse graphics problem has proven to be a challenging quest in the domain of Computer Vision. While many methods have historically emerged (Cohen & Szeliski, 2014; Park et al., 2019; Niemeyer et al., 2020), particularly following the introduction of Neural Radiance Fields (Mildenhall et al., 2020, NeRF), the question has mostly remained unchanged: how to model an object or scene, using only its captured images? While this question continues to be an active area of research, our work targets a scaled version of the original problem. In this paper, we introduce "*scaled* inverse graphics" as the task of concurrently applying inverse graphics over a set of numerous scenes.

We identify *scaled* inverse graphics as an increasingly prominent challenge in recent works. Notably, works tackling 3D generative models (Shue et al., 2023; Müller et al., 2023; Erkoç et al., 2023; Liu et al., 2024) typically require the creation of NeRF datasets, which serve as a prerequisite for training. This is prohibitive, as creating large-scale datasets of implicit scene representations entails significant computational costs. This problem also emerges in practical applications, where efficiently scaling inverse graphics unlocks new ways in which 3D modeling techniques could be leveraged (e.g. modeling product inventories). While this problem has traditionally been tackled in a naive manner where scenes are independently learned, we propose a more efficient alternative to learning scenes in isolation, thereby reducing overall computational costs, without compromising rendering quality.

In this paper, we present a novel technique that addresses scaled inverse graphics. We adopt Tri-Plane representations (Chan et al., 2022b), as they are efficient and lightweight representations that are well-suited for the framework of scaled inverse graphics, and as they have been the primary choice of representations when learning NeRF datasets. Our primary objective is to compact the information required to learn individual scenes when learning large sets of scenes. To this end, we propose a **Micro-Macro decomposition** of Tri-Planes that splits learned features into shared features modeling general information about the scenes in the dataset, and scene-specific features. Concurrently, we learn our scenes in a 3D latent space, thereby alleviating the NeRF rendering bottleneck, and accelerating our training. Our method operates in two stages. **In the first stage,** we train on a subset of scenes the compression model, comprised of the autoencoder and the shared (Macro) Tri-Planes features. **In the second stage,** we utilize these trained components to learn the remaining scenes on smaller representations, thereby reducing the optimization space per scene.

We conduct extensive experiments to evaluate our method in terms of both resource costs and rendering quality when learning large sets scenes, and compare it against the current naive baseline. We further provide an expansive illustration of the resource costs of our method, alongside an ablation study, hence justifying our design choices. As illustrated in Fig. 1, our method presents both the lowest training time and memory footprint in scaled inverse graphics as compared to other methods applied independently on each scene, all while demonstrating comparable NVS quality to Tri-Planes.

A summary of our contribution can be found below:

- We identify the problem of scaled inverse graphics and address it through a novel method,
- We present a novel Micro-Macro decomposition that captures common structures across scenes in shared representations,
- We propose a two-stage training approach that compacts scene representations and enables efficient large-scale scene learning,
- We conduct extensive experiments showing that our method presents the lowest resource costs in scaled inverse graphics while maintaining comparable rendering quality, and justify our design choices through an ablation study

## 2 RELATED WORK

**NeRF resource reductions.** Neural Radiance Fields (Mildenhall et al., 2020, NeRF) achieve impressive performances on the task of Novel View Synthesis (NVS) by adopting a purely implicit representation to model scenes. Following the introduction of NeRFs, several methods have been proposed to improve upon training times and memory costs. Barron et al. (2021; 2022) achieve exceptional quality while requiring low memory capacity to store scenes, as they represent scenes through the weights of neural networks. This however comes with the downside of high training and rendering times due to bottlenecks in volume rendering. To alleviate these issues, some works trade-off compute time for memory usage by explicitly storing proxy features for the emitted radiances and densities in 3D structures (e.g. voxel-based representations (Sun et al., 2022; Chen et al., 2022; Yu et al., 2021; Müller et al., 2022) or plane-based representations (Chan et al., 2022a; Fridovich-Keil et al., 2023; Cao & Johnson, 2023)). Kerbl et al. (2023) and Fridovich-Keil et al. (2022) completely forgo neural networks, achieving real-time rendering but at high memory costs. While previous works have primarily focused on reducing resource costs when learning individual scenes, we propose a method that presents both the lowest training time and memory footprint when learning large sets of scenes, while maintaining rendering quality comparable to that of our base representation. This is partly done by utilizing parts of our pipeline to learn base features that are shared among scenes and

utilized in the second stage of our training, which has proven to be advantageous in previous works (Dupont et al., 2022; Tancik et al., 2021).

**Latent NeRFs.** Latent NeRFs extend NeRFs to render latent image representations in the latent space of an auto-encoder. Several recent work have utilized Latent NeRFs for 3D generation (Metzer et al., 2023; Seo et al., 2023; Ye et al., 2023; Chan et al., 2023), scene editing (Khalid et al., 2023; Park et al., 2024), and scene modeling (Aumentado-Armstrong et al., 2023). However, as latent spaces are not directly compatible with NeRF learning, previous works have resorted to special scene-dependent adaptations, which prevent the concurrent modeling of numerous scenes within a common latent space. In a separate contribution, Anonymous (2024) propose an Inverse Graphics Autoencoder (IG-AE) that embeds a universal 3D-aware latent space compatible with latent NeRF training. While our method is agnostic to the chosen latent space, we build upon the IG-AE architecture to train a 3D-aware latent space that is adapted for scaled inverse graphics, as it currently stands as the only available approach to build NeRF-compatible 3D-aware latent spaces. Accordingly, we adapt the Latent NeRF Training Pipeline to train our decomposed representations in the 3D-aware latent space.

## 3 METHOD

In this section, we present our method for tackling the scaled inverse graphics problem. We start by presenting Tri-Planes (Chan et al., 2022b) and our Micro-Macro Tri-Planes decomposition that allows to compact information. This is done by learning a set of base representations that is shared across scenes (Section 3.1). Next, we present our full training strategy to tackle scaled inverse graphics (Section 3.2). Our approach consists of learning our Micro-Macro decomposed Tri-Planes in a 3D-aware latent space. It operates in two stages. The first stage carries out the computationally intensive task of learning the 3D-aware latent space, while jointly training a subset of scenes and our shared base representations. The second stage benefits from the reduced computational costs enabled after the first stage to learn the remaining scenes.

We denote $\mathcal{S} = \{S_1, ..., S_N\}$ a large set of $N$ scenes drawn from a common distribution. Each scene $S_i = \{(x_{i,j}, p_{i,j})\}_{j=1}^V$ consists of $V$ posed views. Here, $x_{i,j}$ and $p_{i,j}$ respectively denote the $j$-th view and pose of the $i$-th scene $S_i$. We denote $\mathcal{T} = \{T_1, ..., T_N\}$ the set of scene representations modeling the scenes in $\mathcal{S}$. We subdivide $\mathcal{S}$ and $\mathcal{T}$ into two subsets $(\mathcal{S}_1, \mathcal{S}_2)$ and $(\mathcal{T}_1, \mathcal{T}_2)$ at random, respectively containing $N_1$ and $N_2$ scenes, with $N_1 < N_2$.

### 3.1 MICRO-MACRO TRI-PLANES DECOMPOSITION

Tri-Plane representations (Chan et al., 2022a) are explicit-implicit scene representations enabling scene modeling in three axis-aligned orthogonal feature planes, each of resolution $K \times K$ with feature dimension $F$. To query a 3D point $x \in \mathbb{R}^3$, it is projected onto each of the three planes to retrieve bilineraly interpolated feature vectors. These feature vectors are then aggregated via summation and passed into a small neural network with parameters $\alpha$ to retrieve the corresponding color and density, which are then used for volume rendering (Kajiya & Von Herzen, 1984).

We adopt Tri-Plane representations due to their efficient and lightweight architectures, as well as as their widespread use in previous works for constructing NeRF datasets (Shue et al., 2023; Liu et al., 2024). Additionally, the explicit nature of Tri-Planes enables their modularity, an essential property for our Micro-Macro decomposition. While Tri-Planes are traditionally used to model scenes in the RGB space, we utilize them to learn scenes in the latent space of an auto-encoder, defined by an encoder $E_\phi$ and a decoder $D_\psi$. Given a camera pose $p$, we render a latent Tri-Plane $T_i$ as follows:

$$\tilde{z}_{i,j} = R_\alpha(T_i, p_j), \qquad\qquad \tilde{x}_{i,j} = D_\psi(\tilde{z}_{i,j}), \qquad\qquad (1)$$

where $R_\alpha$ is the Tri-Plane renderer with trainable parameters $\alpha$, $\tilde{z}_{i,j}$ is the rendered latent image, and $\tilde{x}_{i,j}$ is the corresponding decoded rendering.

To learn a common structure across our large set of scenes, we introduce a novel approach that splits Tri-Planes into scene-specific features, and features representing global structures. As such, we decompose Tri-Plane representations $T_i$ into "Micro" planes $T_i^{\text{mic}}$ integrating scene specific information, and "Macro" planes $T_i^{\text{mac}}$ that encompass global information, as follows:

$$T_i = T_i^{\text{mic}} \oplus T_i^{\text{mac}}, \qquad\qquad (2)$$

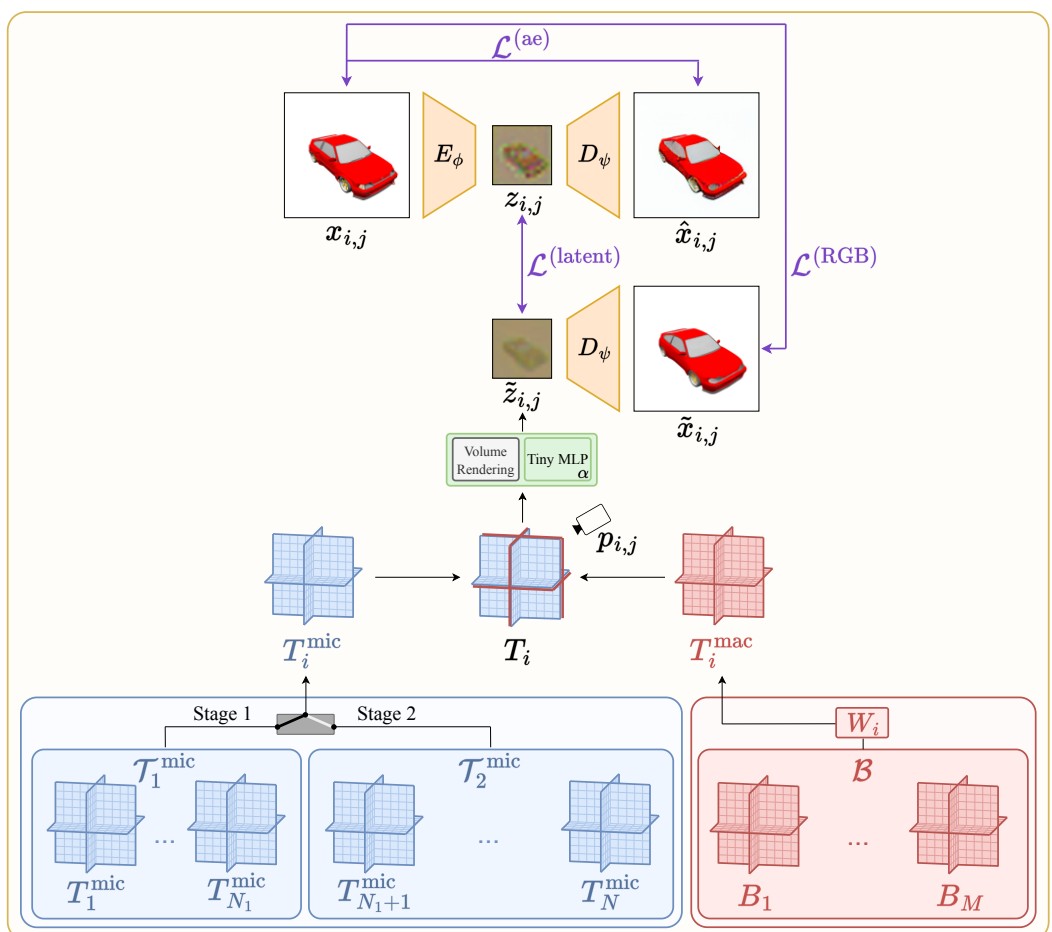

Figure 2: **Learning a large set of scenes.** We learn a large set of scenes using a two-stage approach. Stage 1 jointly learns a small subset of scenes by training the micro-planes $\mathcal{T}_1^{\mathrm{mic}}$, the shared base planes $\mathcal{B}$, the weights $W_i$, as well as the encoder $E_\phi$ and decoder $D_\psi$. Stage 2 learns the rest of the scenes by training $\mathcal{T}_2^{\mathrm{mic}}$ and $W_i$ while fine-tuning $D_\psi$ and $\mathcal{B}$. This stage exclusively uses $\mathcal{L}^{(\mathrm{latent})}$, and then switches to $\mathcal{L}^{(\mathrm{RGB})}$. Note that $T_i^{\mathrm{mac}}$ is computed by a weighted summation over the $M$ shared base planes $\mathcal{B}$, with weights $W_i$.

where $\oplus$ concatenates two Tri-Planes along the feature dimension. We denote by $F^{\mathrm{mic}}$ the number of local features in $T_i^{\mathrm{mic}}$ and by $F^{\mathrm{mac}}$ the number of global features in $T_i^{\mathrm{mac}}$, with the total number of features $F = F^{\mathrm{mic}} + F^{\mathrm{mac}}$.

The micro planes $T_i^{\mathrm{mic}}$ are scene-specific, and are hence independently learned for every scene. The macro planes $T_i^{\mathrm{mac}}$ represent globally captured information that is relevant for the current scene. They are computed for each scene from globally shared Tri-Plane representations $\mathcal{B} = \{B_k\}_{k=1}^M$ by the weighted sum:

$$T_i^{\mathrm{mac}} = W_i B = \sum_{k=1}^M w_i^k B_k \, , \tag{3}$$

where $W_i$ are learned coefficients for scene $S_i$, and $B_k$ are jointly trained with every scene. With this approach, the number of micro planes $N$ scales directly with the number of scenes, while the number of macro planes $M$ is a chosen hyper-parameter. We take $M > 1$ in order to capture diverse information, which our experiments showed to be beneficial for maintaining rendering quality. Overall, our Micro-Macro decomposition allows to accelerate our training and reduce its memory footprint, as we divide the number of trainable features by a factor of $\frac{F}{F^{\mathrm{mic}}}$, asymptotically.

## 3.2 LEARNING A LARGE SET OF 3D SCENES

This section outlines our two-stage training approach to learn a large set of scenes. Fig. 2 provides an overview of our training pipeline. For clarity, the corresponding detailed algorithm is written in Appendix C.

**Stage 1: Learning the latent space and $\mathcal{T}_1$.** The goal in this stage is to train our auto-encoder, while simultaneously learning the representations $\mathcal{T}_1$ modeling the scenes $\mathcal{S}_1$. It is important to note that training the representations $\mathcal{T}_1$ implies training both their scene-specific micro planes, and the globally shared base planes that will also be utilized in the next stage. To learn our latent space and $\mathcal{T}_1$, we implement the 3D regularization losses from Anonymous (2024) recalled below – which could be equivalently replaced by any other 3D-compatible autoencoding method. We supervise a Tri-Plane $T_i$ and the encoder $E_\phi$ in the latent space with the loss $L^{(\text{latent})}$:

$$L_{i,j}^{(\text{latent})}(\phi, T_i, \alpha) = \|z_{i,j} - \tilde{z}_{i,j}\|_2^2 \, , \qquad (4)$$

where $z_{i,j} = E_\phi(x_{i,j})$ is the encoded ground truth image, and $\tilde{z}_{i,j} = R_\alpha(T_i, p_{i,j})$ is the rendered latent image. This loss optimizes the encoder parameters and the Micro-Macro Tri-Plane parameters to align the encoded latent images $z_{i,j}$ and the Tri-Planes rendering $\tilde{z}_{i,j}$. We also supervise $T_i$ and the decoder $D_\psi$ in the RGB space via $L^{(\text{RGB})}$:

$$L_{i,j}^{(\text{RGB})}(\psi, T_i, \alpha) = \|x_{i,j} - \tilde{x}_{i,j}\|_2^2 \, , \qquad (5)$$

where $x_{i,j}$ is the ground truth image, and $\tilde{x}_{i,j} = D_\psi(\tilde{z}_{i,j})$ is the decoded rendering. This loss ensures a good Tri-Plane rendering quality when decoded to the RGB space, and finds the optimal decoder for this task. Finally, we adopt the reconstructive objective $L^{(ae)}$ supervising the auto-encoder:

$$L_{i,j}^{(ae)}(\phi, \psi) = \|x_{i,j} - \hat{x}_{i,j}\|_2^2 \, , \qquad (6)$$

where $\hat{x}_{i,j} = D_\psi(E_\psi(x_{i,j}))$ is the reconstructed ground truth image.

Overall, our full training objective is composed of the three previous losses summed over $\mathcal{S}_1$ to optimize the Micro-Macro Tri-Planes $\mathcal{T}_1$, the encoder $E_\phi$ and the decoder $D_\psi$:

$$\min_{\mathcal{T}_1, \alpha, \phi, \psi} \sum_{i=1}^{N_1} \sum_{j=1}^{V} \lambda^{(\text{latent})} L_{i,j}^{(\text{latent})}(\phi, T_i, \alpha) + \lambda^{(\text{RGB})} L_{i,j}^{(RGB)}(\psi, T_i, \alpha) + \lambda^{(\text{ae})} L_{i,j}^{(ae)}(\phi, \psi) \, , \quad (7)$$

where $\lambda^{(\text{latent})}$, $\lambda^{(\text{RGB})}$, and $\lambda^{(\text{ae})}$ are hyper-parameters. In practice, we start this optimization process with a warm-up stage where the auto-encoder is frozen and only $L^{(\text{latent})}$ is activated. This is done to warm-up the Tri-Planes $\mathcal{T}_1$ and avoid backpropagating random gradients into the auto-encoder.

By the end of this stage, we obtain a custom 3D-aware latent space as well as shared Tri-Planes $\mathcal{B}$ that are specialized on the dataset at hand. These components are passed onto the next stage to allow for an accelerated training of the remaining scenes $\mathcal{T}_2$. These accelerations come from the reduced image resolution enabled by the 3D-aware latent space on the one hand, and the compact scene representations enabled by our Micro-Macro Tri-Planes which reduce the number of trainable features on the other hand.

**Stage 2: Learning $\mathcal{T}_2$.** The goal in this stage is to train the remaining scenes $\mathcal{T}_2$ with alleviated resource costs thanks to the optimizations obtained in the previous stage. To do so, we adapt the Latent NeRF Training Pipeline from Anonymous (2024) to Tri-Planes and scale it via our Micro-Macro decomposition. In this stage, we use the learned autoencoder and the global planes $\mathcal{B}$ from stage 1. In order to relax our optimization objective, we continue to fine-tune the learned global planes. We first optimize the representations $\mathcal{T}_2$ via a Latent Supervision objective as follows:

$$\min_{\mathcal{T}_2, \alpha} \sum_{i=N_1+1}^{N} \sum_{j=1}^{V} L_{i,j}^{(\text{latent})}(\phi, T_i, \alpha) \, . \qquad (8)$$

This objective optimizes the representations in $\mathcal{T}_2$ to reproduce the latent images. Subsequently, we continue with an RGB Alignment which also fine-tunes the decoder for the current scenes:

$$\min_{\mathcal{T}_2, \alpha, \psi} \sum_{i=N_1+1}^{N} \sum_{j=1}^{V} L_{i,j}^{(\text{RGB})}(\psi, T_i, \alpha) \, . \qquad (9)$$

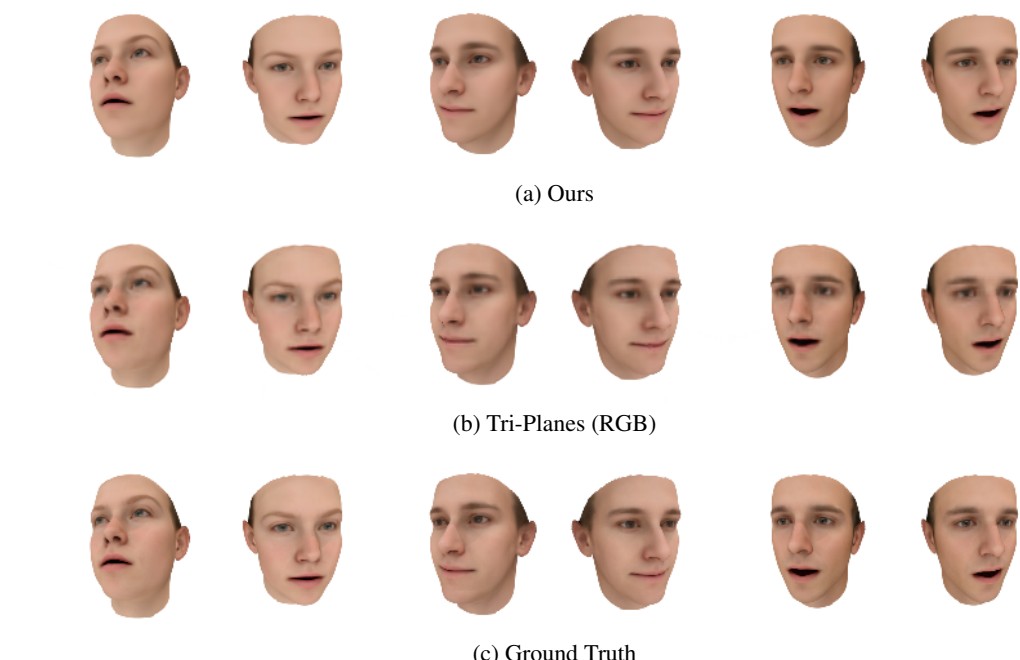

(a) Ours

(b) Tri-Planes (RGB)

(c) Ground Truth

Figure 3: **Qualitative results**. Visual comparison on the Basel Faces dataset of novel view synthesis quality between our method and Tri-Planes (RGB). Our method demonstrates similar rendering quality as compared to Tri-Planes.

Table 1: **Quantitative comparison.** NVS metrics demonstrated by our method and a comparison to standard Tri-Planes. All metrics are averaged over 50 randomly sampled scenes. Our method exhibits NVS quality comparable to that of Tri-Planes.

|  | ShapeNet Cars | | | Basel Faces | | |
|---|---|---|---|---|---|---|
|  | PSNR↑ | SSIM↑ | LPIPS↓ | PSNR↑ | SSIM↑ | LPIPS↓ |
| Tri-Planes (RGB) | 28.56 | **0.9512** | **0.0346** | 36.44 | **0.9791** | 0.0127 |
| Ours | **28.64** | 0.9498 | 0.0367 | **36.82** | 0.9706 | **0.0057** |

The end of this stage marks the end of our training where all the scenes in $\mathcal{T} = \mathcal{T}_1 \cup \mathcal{T}_2$ are now learned. Note that the trained components of our pipeline can still be utilized after this training to learn additional scenes with alleviated resource costs.

## 4 EXPERIMENTS

We assess our method by employing it for the task of scaled inverse graphics. We utilize our method to learn two distinct large-scale datasets: ShapeNet Cars and Basel Faces. For each case, we start by training our method on a subset of scenes, and then utilize it to train the remaining scenes. We evaluate the rendering quality and resource costs of our method and compare it to our base representation. Moreover, we provide a comparison of our resource costs with recent methods when trained independently in large-scale settings. Finally, we present an ablation study to assess the added value of each element of our pipeline.

**Dataset.** We evaluate our method on two datasets: the Cars dataset from ShapeNet (Chang et al., 2015) and the front-facing Basel-Face dataset (Walker et al., 2018). For each dataset, each scene $S_i$ is rendered at a $128 \times 128$ resolution. We take $V = 160$ views for cars, sampled from the upper hemisphere surrounding the object. For faces, we take $V = 50$ front-facing views. In all experiments,

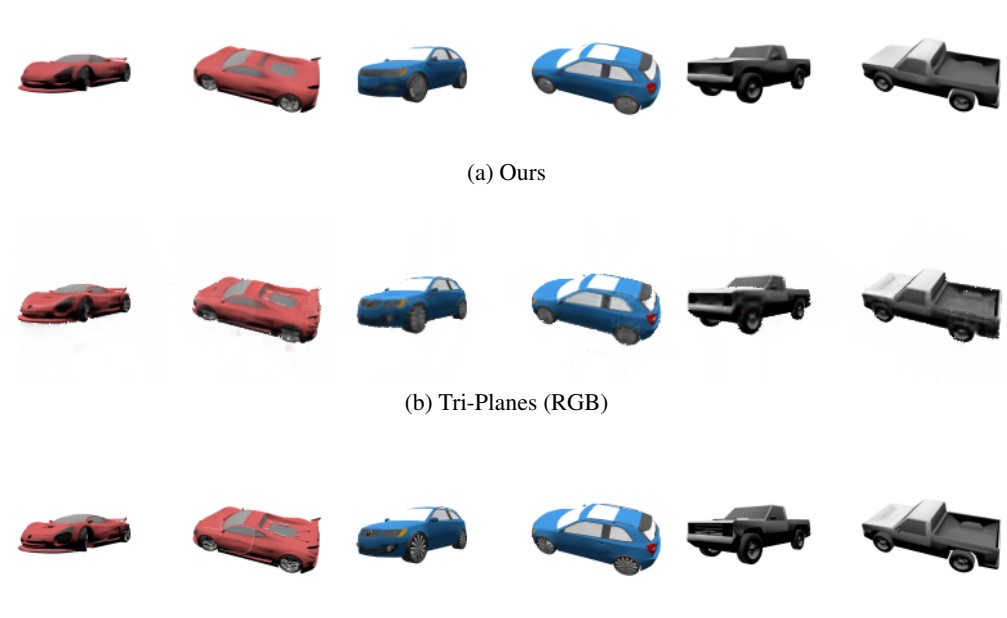

(a) Ours

(b) Tri-Planes (RGB)

(c) Ground Truth

Figure 4: **Qualitative results**. Visual comparison on the ShapeNet Cats dataset of novel view synthesis quality between our method and Tri-Planes (RGB). Our method demonstrates similar rendering quality as compared to Tri-Planes.

we take $90\%$ of views for training and $10\%$ for testing. The test views are reserved to evaluate the NVS performances of Tri-Planes.

**Implementation details.** For all experiments, we learn the 3D-aware latent space on $N_1 = 500$ scenes. Then, we utilise it to learn $N_2 = 1500$ scenes in the second phase. We take $F^{\mathrm{mic}} = 10$, $F^{\mathrm{mac}} = 22$, and $M = 50$. We detail our hyper-parameters in Appendix D. We adopt the pre-trained VAE from Stable Diffusion (Rombach et al., 2022). Our training is done on $4\times$ NVIDIA L4 GPUs. Our losses are computed on mini-batches of 32 images. Our code is available in the supplementary material and will be open-sourced upon publication.

## 4.1 EVALUATIONS

In this section, we detail our evaluation scheme to assess the NVS quality and the resource costs in terms of training time and memory footprint.

**NVS Quality.** To evaluate the NVS quality of the learned scenes $\mathcal{T}$, we compute the Peak Signal-to-Noise Ratio (PSNR ↑), the Strutural Similarity Index Measure (SSIM ↑) and the Learned Perceptual Image Patch Similarity (Zhang et al., 2018, LPIPS ↓) on never-seen test views. Table 1 and Figs. 3 and 4 illustrate our quantitative and qualitative results. We compare our results with a classical training of Tri-Planes in the image space, denoted "Tri-Planes (RGB)". Our method achieves similar NVS quality as compared to Tri-Planes (RGB). For a fair comparison, we use the same plane resolutions $K = 64$ and the same number of plane features $F = 32$ in all our experiments. All methods are trained until convergence. Note that, due to the long training times of Tri-Planes (RGB), we carry out our comparison on a subset of 50 scenes randomly sampled from $\mathcal{S}$. Furthermore, we present in Appendix B a comparison of the NVS quality of our method between stages 1 and 2. Both stages exhibit similar NVS performances.

**Time costs.** As presented, our method starts by jointly training the autoencoder and $N_1$ scenes, and then utilizes the trained autoencoder for the remaining $N_2 = N - N_1$ scenes. For $N \geq N_1$, our

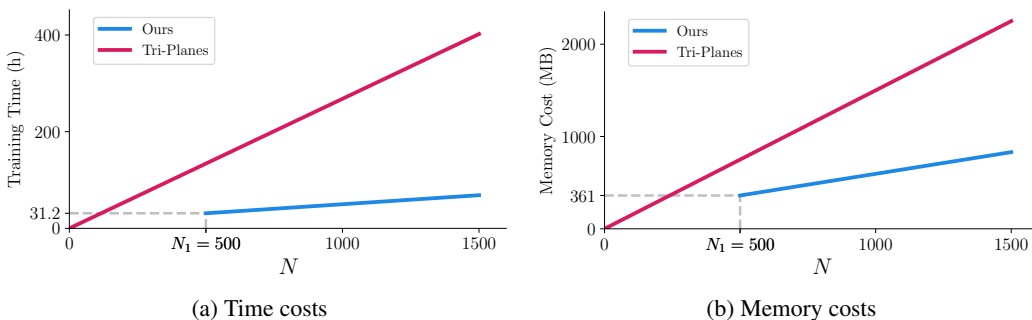

(a) Time costs         (b) Memory costs

Figure 5: **Resource costs comparison.** Comparison of the training time and memory costs required by our method and classic Tri-Planes when scaling the number of training scenes $N$ in the ShapeNet Cars dataset. Our method demonstrates more favorable scalability as $N$ increases.

Table 2: **Resource Costs.** Comparison of per-scene resource costs in scaled inverse graphics between our method and independently trained RGB Tri-Planes on ShapeNet Cars. Our method presents significantly alleviated per-scene resource costs following the first stage of our training.

|  | $\tau$ (min) | $\mu$ (MB) | Rendering Time (ms) | Decoding Time (ms) |
|---|---|---|---|---|
| Tri-Planes (RGB) | 16.02 | 1.50 | 23.30 | 0 |
| Our method | 2.23 | 0.48 | 0.36 | 9.71 |

total training time is written as:

$$t^{(\text{ours})}(N) = t_1 + (N - N_1)\tau^{(\text{ours})} \,, \tag{10}$$

where $t_1$ is the time required in the first stage of our training and $\tau^{(\text{ours})}$ is the training time per scene in our second phase.

We also denote by $t^{(\text{rgb})}(N)$ the time required to learn $N$ scenes with regular Tri-Planes independently trained on RGB images:

$$t^{(\text{rgb})}(N) = N\tau^{(\text{rgb})} \,, \tag{11}$$

where $\tau^{(\text{rgb})}$ is the training time per RGB scene for Tri-Planes.

**Memory costs.** We denote $m_1$ the memory cost to store the components of the first phase of our training (i.e. the encoder, decoder, global Tri-Planes, micro Tri-Planes, and corresponding learned coefficients). Our total memory footprint when learning $N \geq N_1$ scenes is written as:

$$m^{(\text{ours})}(N) = m_1 + (N - N_1)\mu^{(\text{ours})} \,, \tag{12}$$

where $\mu^{(\text{ours})}$ is the memory cost to store one scene (i.e. a micro plane and the learned macro coefficients) in the second stage. We also denote by $m^{(\text{rgb})}(N)$ the memory cost to store $N$ scenes with regular Tri-Planes:

$$m^{(\text{rgb})}(N) = N\mu^{(\text{rgb})} \,, \tag{13}$$

where $\mu^{(\text{rgb})}$ is the memory cost to store one RGB Tri-Plane.

The first stage of our training necessitates $t_1 = 31.2$ hours and $m_1 = 361$ MB when training $N_1 = 500$ scenes. Table 2 details our resource costs in the second stage. When $N$ is large, our method asymptotically reduces the training time required to learn individual scenes by 86% and memory costs by 68%. Moreover, rendering using our method requires 98% less time. While this is followed by a decoding time of 9.61 ms, producing an RGB image using our method overall requires 56% less time. Fig. 5 illustrates the evolution of the training time and memory cost of our method and our baseline as $N$ grows. Our method demonstrates favorable scaling when learning a large number

Table 3: **Ablation Study.** Quantitative results of our ablation study. NVS metrics are computed on the same $50$ randomly sampled scenes from the ShapeNet Cars dataset. Our method presents similar NVS performances to Tri-Planes, while outperforming our ablations.

| | Latent Space | Micro Planes | Macro Planes | PSNR↑ | SSIM↑ | $\tau$ (min) | $\mu$ (MB) |
|---|---|---|---|---|---|---|---|
| Ours-Micro | ✓ | ✓ | ✗ | 27.64 | 0.9409 | 3.21 | 1.50 |
| Ours-Macro | ✓ | ✗ | ✓ | 27.51 | 0.9346 | 1.79 | 0.0008 |
| Ours-$M = 1$ | ✓ | ✓ | ✓ | 27.69 | 0.9416 | 2.12 | 0.48 |
| Ours-RGB | ✗ | ✓ | ✓ | 27.71 | 0.9418 | 15.88 | 0.48 |
| Tri-Planes (RGB) | ✗ | ✓ | ✗ | 28.56 | 0.9512 | 16.02 | 1.50 |
| Ours | ✓ | ✓ | ✓ | 28.64 | 0.9498 | 2,23 | 0.48 |

of scenes. However, it is only accessible after training the first set of $N_1$ scenes. Reducing $N_1$ would allow leveraging our alleviated resource cost earlier, which we see as a direction of future work.

Fig. 1 provides an overview comparison of our method with recent methods when used for scaled inverse graphics. Our approach demonstrates the lowest resource costs in both training time and memory footprint, while maintaining a comparable NVS quality to Tri-Planes. The data associated with this figure can be found in Appendix A.

## 4.2 ABLATIONS

To justify our choices and explore further, we present an ablation study of our method, for which the results are presented in Table 3. The first ablation, "**Ours-Micro**", eliminates the Micro-Macro decomposition, and consequently global information sharing (i.e. $F^{\mathrm{mac}} = 0, F^{mic} = F$). This ablation showcases a slight degradation of quality as compared to our full method, but more importantly, it would result in higher resource costs as it eliminates the shared base representations and requires more learnable features per scene. The second ablation, "**Ours-Macro**", eliminates local features from Tri-Planes and relies only on global features (i.e. $F^{mic} = 0, F^{\mathrm{mac}} = F$). This setting also showcases a degradation in NVS quality as compared to our method, as it only relies on shared planes to represent individual scenes. The third ablation, "**Ours-$M = 1$**" reduces the set of shared planes $\mathcal{B}$ to one Tri-Plane. This ablation demonstrates NVS performances that are slightly higher than Ours-Micro, but still lower than our method, highlighting the necessity for *a set* of global planes. The fourth ablation, "**Ours-RGB**" ablates the latent space and trains Micro-Macro decomposed Tri-Planes in the RGB space. It also presents decreased performances as compared to our method, and thus highlighting the advantage of doing our Micro-Macro decomposition on latent scenes. Note that ablating the latent space as well as information sharing is equivalent to the vanilla "**Tri-Planes (RGB)**" setting, which presents a comparable rendering quality with respect to our method, with significantly higher resource costs.

## 5 CONCLUSION

In this paper, we introduce *scaled* inverse graphics and recognize the necessity for methods that efficiently tackle this problem. We propose a novel method that learns scenes in a custom 3D-aware latent space, and uses a novel Micro-Macro Tri-Plane decomposition that compacts the representation of individual scenes by adopting a set of shared representations. Our method demonstrates significantly lower training time and memory costs in scaled inverse graphics as compared to recent methods, while maintaining a comparable rendering quality. We consider this work to be an initial step in the direction of efficiently scaling inverse graphics methods.

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

## A  RESOURCE COSTS COMPARISON

For transparency, Table 4 presents the data behind Fig. 1.

Table 4: **Resource Costs Comparison.** This table presents the underlying data behind Fig. 1. Our method showcases the lowest training time and memory costs when training $N = 2000$ scenes, and a comparable NVS quality to Tri-Planes, our base representation. All models are trained on $4\times$ NVIDIA L4 GPUs.

|  | Training Time (min $\times 10^3$) | Memory Footprint (MB $\times 10^3$) | PSNR↑ |
|---|---|---|---|
| Vanilla-NeRF | 318.40 | 25.14 | 40.16 |
| Instant-NGP | 5.50 | 359.40 | 37.71 |
| TensoRF | 34.47 | 397.78 | 40.38 |
| K-Planes | 37.68 | 801.48 | 32.59 |
| Tri-Planes | 32.04 | 3.00 | 30.03 |
| Ours | 5.22 | 1.08 | 31.20 |

## B  SUPPLEMENTARY RESULTS

Tables 5 and 6 present our NVS metrics in both stages of our training for the ShapeNet Cars and the Basel Faces datasets, respectively. All metrics are computed on never-seen test views from 50 randomly sampled scenes from each dataset. Both stages of our training present similar NVS performances.

Table 5: **Quantitative comparison.** NVS performances on ShapeNet Cars in both stages of our training.

|  | ShapeNet Cars | | | | | |
|---|---|---|---|---|---|---|
|  | $\mathcal{S}_1$ | | | $\mathcal{S}_2$ | | |
|  | PSNR↑ | SSIM↑ | LPIPS↓ | PSNR↑ | SSIM↑ | LPIPS↓ |
| Tri-Planes (RGB) | 28.49 | 0.9539 | 0.0291 | 28.58 | 0.9505 | 0.0360 |
| Ours | 28.14 | 0.9505 | 0.0301 | 28.77 | 0.9496 | 0.0383 |

Table 6: **Quantitative comparison.** NVS performances on Basel Faces in both stages of our training.

|  | Basel Faces | | | | | |
|---|---|---|---|---|---|---|
|  | $\mathcal{S}_1$ | | | $\mathcal{S}_2$ | | |
|  | PSNR↑ | SSIM↑ | LPIPS↓ | PSNR↑ | SSIM↑ | LPIPS↓ |
| Tri-Planes (RGB) | 36.82 | 0.9807 | 0.0122 | 36.35 | 0.9787 | 0.0129 |
| Ours | 36.17 | 0.9678 | 0.0062 | 36.99 | 0.9712 | 0.0056 |

## C  TWO-STAGE TRAINING ALGORITHM

For clarity, Algorithm 1 provides a detailed outline of our two-stage training method.

## D  HYPERPARAMETERS

For reproducibility purposes, Tables 7 and 8 expose our hyperparameter settings respectively for stage 1 and stage 2 of our training. A more detailed list of our hyperparameters can be found in the configuration files of our open-source code.

---

**Algorithm 1:** Training a large set of scenes.

---

**Input:** $\mathcal{S}_1, \mathcal{S}_2, N_1, N_2, V, E_\phi, D_\psi, \mathcal{R}_\alpha, N_{\text{epoch}}^{(1)}, N_{\text{epoch}}^{(2)}, N_{\text{epoch}}^{(\text{LS})}, \lambda^{(\text{latent})}, \lambda^{(\text{RGB})}, \lambda^{(\text{ae})}$,
optimizer

**Random initialization:** $\mathcal{T}_1^{\text{mic}}, \mathcal{T}_2^{\text{mic}}, W, \mathcal{B}$

```
// Stage 1
```
**1** **for** $N_{\text{epoch}}^{(1)}$ *steps* **do**

**2**    **for** $(i,j)$ *in* $\text{shuffle}(\llbracket 1, N_1 \rrbracket \times \llbracket 1, V \rrbracket)$ **do**

      `// Compute Micro-Macro decomposition`

**3**       $T_i^{(\text{mic})}, T_i^{(\text{mac})} \leftarrow \mathcal{T}_1^{(\text{mic})}[i], W_i \mathcal{B}$

**4**       $T_i \leftarrow T_i^{(\text{mic})} \oplus T_i^{(\text{mac})}$

      `// Encode, Render & Decode`

**5**       $x_{i,j}, p_{i,j} \leftarrow \mathcal{S}_1[i][j]$

**6**       $z_{i,j} \leftarrow E_\phi(x_{i,j})$

**7**       $\tilde{z}_{i,j} \leftarrow \mathcal{R}_\alpha(T_i, p_{i,j})$

**8**       $\hat{x}_{i,j} \leftarrow D_\psi(z_{i,j})$

**9**       $\tilde{x}_{i,j} \leftarrow D_\psi(\tilde{z}_{i,j})$

      `// Compute losses`

**10**      $L_{i,j}^{(\text{latent})} \leftarrow \|z_{i,j} - \tilde{z}_{i,j}\|_2^2$

**11**      $L_{i,j}^{(\text{RGB})} \leftarrow \|x_{i,j} - \tilde{x}_{i,j}\|_2^2$

**12**      $L_{i,j}^{(ae)} \leftarrow \|x_{i,j} - \hat{x}_{i,j}\|_2^2$

**13**      $L_{i,j} \leftarrow \lambda^{(\text{latent})} L_{i,j}^{(latent)} + \lambda^{(\text{RGB})} L_{i,j}^{(RGB)} + \lambda^{(\text{ae})} L_{i,j}^{(ae)}$

      `// Backpropagate`

**14**      $T_i^{(\text{mic})}, W_i, \mathcal{B}, \alpha, \phi, \psi \leftarrow \text{optimizer.step}(L_{i,j})$

**15**

```
// Stage 2
```
**16** $\text{epoch} = 1$

**17** **for** $N_{\text{epoch}}^{(2)}$ *steps* **do**

**18**    **for** $(i,j)$ *in* $\text{shuffle}(\llbracket N_1 + 1, N_1 + N_2 \rrbracket \times \llbracket 1, V \rrbracket)$ **do**

      `// Compute Micro-Macro decomposition`

**19**      $T_i^{(\text{mic})}, T_i^{(\text{mac})} \leftarrow \mathcal{T}_2^{(\text{mic})}[i], W_i \mathcal{B}$

**20**      $T_i \leftarrow T_i^{(\text{mic})} \oplus T_i^{(\text{mac})}$

      `// Encode, Render & Decode`

**21**      $x_{i,j}, p_{i,j} \leftarrow \mathcal{S}_2[i][j]$

**22**      $z_{i,j} \leftarrow E_\phi(x_{i,j})$

**23**      $\tilde{z}_{i,j} \leftarrow \mathcal{R}_\alpha(T_i, p_{i,j})$

**24**      $\tilde{x}_{i,j} \leftarrow D_\psi(\tilde{z}_{i,j})$

**25**

**26**      **if** $\text{epoch} \leq N_{\text{epoch}}^{(\text{LS})}$ **then**

       `// Latent Supervision`

**27**        $L_{i,j}^{(\text{latent})} \leftarrow \|z_{i,j} - \tilde{z}_{i,j}\|_2^2$

**28**        $T_i^{(\text{mic})}, W_i, \mathcal{B}, \alpha \leftarrow \text{optimizer.step}(L_{i,j}^{(\text{latent})})$

**29**      **else**

       `// RGB Alignment`

**30**        $L_{i,j}^{(\text{RGB})} \leftarrow \|x_{i,j} - \tilde{x}_{i,j}\|_2^2$

**31**        $T_i^{(\text{mic})}, W_i, \mathcal{B}, \alpha, \psi \leftarrow \text{optimizer.step}(L_{i,j}^{(\text{RGB})})$

**32**    $\text{epoch} \leftarrow \text{epoch} + 1$

---

Table 7: **Stage 1 hyperparameters.**

| Parameter | Value |
|---|---|
| General | |
| Number of scenes $N_1$ | 500 |
| Pretraining epochs | 50 |
| Training epochs | 50 |
| Tri-Planes | |
| Number of micro feature $F_{\text{mic}}$ | 10 |
| Number of macro feature $F_{\text{mac}}$ | 22 |
| Number of base plane $M$ | 50 |
| Tri-Planes resolution | 64 |
| Loss | |
| $\lambda^{(\text{latent})}$ | 1 |
| $\lambda^{(\text{RGB})}$ | 1 |
| $\lambda^{(\text{ae})}$ | 0.1 |
| Optimization (warm-up) | |
| Optimizer | Adam |
| Batch size | 512 |
| Learning rate (Tri-Planes $T_i^{(\text{mic})}$) | $10^{-2}$ |
| Learning rate (Triplane renderer $R_\alpha$) | $10^{-2}$ |
| Learning rate (Coefficients $w_i^k$) | $10^{-2}$ |
| Learning rate (Base planes $B_k$) | $10^{-2}$ |
| Scheduler | Multistep |
| Decay factor | 0.3 |
| Decay milestones | $[20, 40]$ |
| Optimization (training) | |
| Optimizer | Adam |
| Batch size | 32 |
| Learning rate (encoder) | $10^{-4}$ |
| Learning rate (decoder) | $10^{-4}$ |
| Learning rate (Tri-Planes $T_i^{(\text{mic})}$) | $10^{-4}$ |
| Learning rate (Triplane renderer $R_\alpha$) | $10^{-4}$ |
| Learning rate (Coefficients $w_i^k$) | $10^{-2}$ |
| Learning rate (Base planes $B_k$) | $10^{-2}$ |
| Scheduler | Multistep |
| Decay factor | 0.3 |
| Decay milestones | $[20, 40]$ |

Table 8: **Stage 2 hyperparameters.**

| Parameter | Value |
|---|---|
| General | |
| Number of scenes $N_2$ | 1500 |
| Number of Latent Supervision epochs $N_{\text{epoch}}^{(\text{LS})}$ | 30 |
| Number of RGB Alignment epochs $N_{\text{epoch}}^{(\text{RA})}$ | 50 |
| Tri-Planes | |
| Number of micro feature $F_{\text{mic}}$ | 10 |
| Number of macro feature $F_{\text{mac}}$ | 22 |
| Number of base plane $M$ | 50 |
| Tri-Planes resolution | 64 |
| Loss | |
| $\lambda^{(\text{latent})}$ | 1 |
| $\lambda^{(\text{RGB})}$ | 1 |
| Optimization (Latent Supervision) | |
| Optimizer | Adam |
| Batch size | 32 |
| Learning rate (Tri-Planes $T_i^{(\text{mic})}$) | $10^{-2}$ |
| Learning rate (Triplane renderer $R_\alpha$) | $10^{-2}$ |
| Learning rate (Coefficients $w_i^k$) | $10^{-2}$ |
| Learning rate (Base planes $B_k$) | $10^{-2}$ |
| Scheduler | Exponential decay |
| Decay factor | 0.941 |
| Optimization (RGB Alignment) | |
| Optimizer | Adam |
| Batch size | 32 |
| Learning rate (decoder) | $10^{-4}$ |
| Learning rate (Tri-Planes $T_i^{(\text{mic})}$) | $10^{-3}$ |
| Learning rate (Triplane renderer $R_\alpha$) | $10^{-3}$ |
| Learning rate (coefficients $w_i^k$) | $10^{-2}$ |
| Learning rate (nase planes $B_k$) | $10^{-2}$ |
| Scheduler | Exponential decay |
| Decay factor | 0.941 |

