# OpenReview forum: "Scaled Inverse Graphics: Efficiently Learning Large Sets of 3D Scenes"
_ICLR.cc/2025/Conference — Submitted to ICLR 2025_

### Official Review · Reviewer_M83u · 2024-10-28

**Soundness:** 2
**Presentation:** 2
**Contribution:** 2
**Rating:** 3
**Confidence:** 3

**Summary:**

The paper deals with the concurrent reconstruction of multiple objects. It bases it works on Triplanes and strikes a balance among training cost and quality.

**Strengths:**

1. Originality. The proposed two-stage training pipeline and micro-macro tri-planes decomposition serves as a novel technique for the concurrent reconstruction of large sets of object

**Weaknesses:**

1. As a submission to ICLR25, it seems unreasonable to only compare quality with NeRF-based methods.  The recent 3DGS has shown significant improvement in training cost and gained a series of improvements. It's beneficial to compare the proposed method with recent FlashGS or gsplat.
2. The comparison with Triplanes seems to be unfair as well. The proposed method indeed increases the model capacity. For an apple-to-apple comparison, the capacity of Triplanes should be better aligned.
3. The ablation is insufficient. What quantitative benefit will be brought by second-stage training over direct training on all scenes? How will the result be influenced if I change the ratio of data assigned to two stages? What about adding more stages?

**Questions:**

1. The writing and organization of the Introduction require improvement. Why is this problem critical? What are typical and critical application scenarios of the problem?
2. The comparison and ablation experiment should be refined, as detailed in Weakness section.

---

> ### Author Response · Authors · 2024-11-28
>
> We thank the reviewer for their constructive remarks.
>
> ### w.1 Baselines.
> We thank the reviewer for this suggestion. We compare with other NeRF variants (Instant-NGP, TensoRF, K-Planes, and Vanilla NeRF in figure 1 and table 4), but indeed lack GS. We will include it in a future version of our paper.
>
>
> ### w.2 Triplane capacity
> We respectfully disagree. Both settings yield a Tri-Plane representation with similar shape (3 x 64 x 64 x 32). Our proposed method actually reduces the overall model capacity, due to the decomposition in the base of global Tri-Planes (shared across all scenes), which leads to fewer trainable parameters overall.
>
> ### w.3 / q.2 Comparing first and second stages
> > What quantitative benefit will be brought by second-stage training over direct training on all scenes?
>
> The second stage yields a faster per-scene optimization than the first stage (around two times faster), since the autoencoder does not need training in this stage.
>
> > How will the result be influenced if I change the ratio of data assigned to two stages?
>
> We thank the reviewer for this interesting suggestion, which reviewer fgoE also pointed out. We will discuss this point in a future version of our paper.
>
> > What about adding more stages?
>
> Adding more stages would not be useful. It would be equivalent to performing the second stage with more scenes.

---

> > ### Comment · Reviewer_M83u · 2024-11-29
> >
> > Blank promise with no solid evidence. The score will not be changed.

---

### Official Review · Reviewer_fgoE · 2024-11-04

**Soundness:** 3
**Presentation:** 2
**Contribution:** 3
**Rating:** 5
**Confidence:** 5

**Summary:**

This paper proposed to address the problem of nerf training for a large set of scenes. In particular, they propose to learn the Micro-macro Tri-planes, where the Macro Tri-planes capture the shared features for the scene and Micro Tri-planes captures the features for each scene. The Nerf is trained on a subset of scenes first and then fine-tuned on remaining scenes. The proposed framework is evaluated on Cars dataset from ShapeNet and front-facing Basel-Face dataset which shows similar rendering performance but reduced computational time and memory cost.

**Strengths:**

The proposed training strategy highly reduced the computational time and memory for nerf training using decouple Tri-plane representations- Macro and Micro Tri-planes, for large set of 3D scenes.

**Weaknesses:**

-	Scene VS Object. The method claimed efficient learning for a large set of 3D scene but the evaluations are done only on objects. No extra experimental results are listed in the supplementary material.
-	Clarification about micro and macro decomposition of Tri-Planes. The current experiments cannot explain whether the Macro Tri-planes capture the global feature. Actually, it is not clear what this global feature means. It would be great to render the image or 3D model based using learned Macro Tri-planes only.
-	Validate the influence on the performance by the ratio of the number of scenes used for training in the first stage and the second stage.

-	Some details are not clearly presented in the paper.

1)	“S describes N scenes drawn from a common distribution.” (line 134-135) It would be great that this definition to be more specific. Are those scenes required to be from the same category or the scene type, such as living room, or kitchen?
2)	Line135-136, the definition of x_{i,j} and p_{i,j} are unclear. I would assume x_{i,j} represents the image rendered at ith view where the camera pose is defined as p_{i,j}. The current definitions are not clear at all.
3)	Line 230-231: Please clarify the encoded ground truth image and latent image. Do you mean the latents of the ground truth image and latent image?

**Questions:**

Overall, the idea is interesting. However, some claims are not supported by the experiments in the paper.
1) insufficient experiments. Will it be good to train the model for the real scene dataset such as ScanNet? We knew that for typical scenes, there are a large number of combination of object to form a scene. It is not clear how this method would perform for scenes. The current experiments didn't fully demonstrate this.

 2) Please also demonstrate what is actually learned by the Macro tri-plane. It would be interesting to show rendered images or geometries.

3) Please provide the relationship of the performance related to the split of scenes for training.

**Details Of Ethics Concerns:**

No concerns.

---

> ### Author Response · Authors · 2024-11-28
>
> We thank the reviewer for their constructive remarks.
>
> ### w.1 / q.1 Scene VS Object.
> As Tri-Planes are usually trained on objects, we follow the same strategy. We will update the title to better reflect our claim.
>
> Thank you for suggesting ScanNet which we might consider in a future version of our paper.
>
> ### w.2 / q.2 Visualization of global Tri-Planes.
> Thank you for your suggestion about adding visualizations of the global Tri-Planes. It is technically complicated to visualize, since global Tri-Planes only capture a part on the scene information and need to be associated with Micro Tri-Planes. We will consider making such visualizations, or other ways to better describe global Tri-Planes, in a future version of our paper.
>
> ### w.3 / q.3 Stage 1 v. stage 2 Scenes ratio
> We thank the reviewer for this interesting suggestion, which reviewer M83u also pointed out.  We will discuss this point in a future version of our paper.

---

### Official Review · Reviewer_86Z6 · 2024-11-05

**Soundness:** 3
**Presentation:** 3
**Contribution:** 3
**Rating:** 5
**Confidence:** 3

**Summary:**

This paper attempts to scale neural representations of scenes, focusing on Tri-planes. Their idea is to learn a global tri-plane that is queried for all new scenes (macro), as well as per-scene tri-planes. When volume-rendering a tri-plane representation, both the macro and scene-specific micro representations are rendered, and features from both are concatenated along the feature dimension. The Macro planes are actually a collection of triplanes, and the final features from the macro planes are a weighted average. The paper shows that the overall amortized cost of the macro-micro plane approach is less than the cost of training a new independent tri-plane representation that does not benefit from having seen other scenes.

**Strengths:**

*I think this is a useful paper that brings scalability to tri-planes. Having the macro-micro decomposition is clearly more efficient than learning one tri-plane per scene.
*Method is clearly explained

**Weaknesses:**

*The classic tri-plane representation (EG3D) involves a single neural network generating the tri-planes from a latent code that are then rendered. Did the authors explore a single network to generate the tri-planes per scene and comparing that cost with the macro-micro approach? I believe at a minimum the authors should perform an ablation by training a single EG3D on their dataset (perhaps with varying feature dimensionality to match the macro-micro feature space).

**Questions:**

Line 134-138: why does the data need to be divided into two disjoint subsets?

---

> ### Author Response · Authors · 2024-11-28
>
> We thank the reviewer for their constructive remarks.
>
> ### w: Comparison with EG3D
> We only take the proposed representation (Tri-Planes) from EG3D, and not the generative model.
> Tri-Planes have become widely used in the literature after EG3D, and have been especially adopted for creating NeRF datasets.
> In this work, we are not working on a generative model and as such we don't propose one, and don't compare with the one in EG3D.
>
> ### q: Disjoint subsets
> To learn a large set of NeRFs, we divide it into two disjoint subsets.
> The first subset is learned in the first stage, and the second subset is learned in the second stage.
> If the subsets were not disjoint, the same scene would be learned twice, which would unnecessarily consume computational resources.

---

### Official Review · Reviewer_8Gu4 · 2024-11-07

**Soundness:** 2
**Presentation:** 3
**Contribution:** 1
**Rating:** 1
**Confidence:** 5

**Summary:**

The paper considers the task of distilling knowledge from a large set of 3D scenes to improve resource use at inference time. Specifically, the paper proposes to learn a tri-plane representation using an auto-encoder. The latent tri-plane is further decomposed into a global tri-plane shared across all scenes in a category and a scene-specific tri-plane. The paper’s baseline is a scene-specific tri-plane, and experiments are conducted on ShapeNet Cars and Basel Faces. The paper demonstrates that after the distillation phase, their method is significantly faster and requires fewer parameters than per-scene tri-plane while achieving similar performance.

**Strengths:**

* The paper is easy to read and understand. The methodology is clear and well-explained
* The explicit decomposition of shared global embeddings and scene-specific embeddings in a concatenated Tri-Plane space is interesting.

**Weaknesses:**

* Lack of compelling baselines. Although the paper presents a latent NeRF, there are no latent NeRF baselines. The paper is only compared to a per-scene Tri-NeRF. Some relevant NeRF-centric papers that learn a shared representation that is then used to quickly adapt to new scenes include [1] and [2]. Arguably, any single-view or multi-view feedforward method trained using analysis-by-synthesis rendering losses (MVSplat, Pixelsplat, Splatter Image, CAT3D, SparseFusion, SRT, UP-SRT, RUST, etc) is also trained on a large set of 3D scenes and then adapted quickly to an inference scene. These would still fit into the proposed task setting and would likely be several orders of magnitude faster than the proposed method (miliseconds vs 2 min). Finally, since the method still uses 144 input views for shapenet cars and 45 for faces, any recent fast scene-specific method would perform better than Tri-Planes and be faster (a few seconds for 3DGS and InstantNGP). Yet, even with the distillation, the proposed method is not much better than the Tri-Planes baseline, performing slightly worse in LPIPS for shapenet cars.
* Weak motivation. The primary problem of reducing resource costs at inference time seems to be better solved using existing fast (3DGS, INGP) and low memory (NeRFLight) NeRFs or even distillation into smaller NeRFs (Plenoctrees). These solutions would be orders of magnitudes faster (few seconds vs 31 hours + 2.2min/scene) and lower memory (10-30MB vs 360 MB). The notable challenge described in the introduction is that “the creation of NeRF datasets, which serve as a prerequisite for training… is prohibitive, as creating large-scale datasets of implicit scene representations entails significant computational costs” [L55-58]. This problem does not seem to be addressed by this paper since it still requires constructing a large dataset for pre-training Stage 1.
* Memory costs are misleading. The paper claims that decomposing global and scene-specific parameters reduces memory footprint because it reduces trainable features by F/F_mic [L214-215]. However, this doesn’t make sense because the global features must still be stored in memory in order to decode the scene. There doesn’t seem to be any memory savings from the core technical contribution of the paper.

[1] Tancik et al. Learned Initializations for Optimizing Coordinate-Based Neural Representations.

[2] Hamdi et al. SPARF: Large-Scale Learning of 3D Sparse Radiance Fields from Few Input Images.

**Questions:**

What does it mean for IG-AE to be the “only available approach to build NeRF-compatible 3D-aware latent spaces” [L120-121]?

---

> ### Author Response · Authors · 2024-11-28
>
> We thank the reviewer for their constructive remarks.
>
> ### w.1: Baselines
> While it is true that the paper does not compare latent NeRFs to other latent NeRF baseline, **such baseline does not exist in the context of scene reconstruction**. However, we do compare to other NeRF representations (Instant-NGP, TensoRF, K-Planes, and Vanilla NeRF (figure 1, table 4)).
> We refer the reviewer to our global answer "Comparison with feedforward baselines", [here](https://openreview.net/forum?id=GSckuQMzBG&noteId=L9CjDTGnsr).
>
> ### w.2: Motivation and comparisons
> Methods like INGP and 3DGS are fast but exhibit high memory footprint.
> We compare our method with INGP in figure 1 and table 4, and show that our method is both faster and has lower memory footprint.
>
> The dataset used in the first stage is a subset from scenes to be learned. The first stage is not a pre-training stage. It is the first stage of our training and yields 500 trained Tri-Planes.
> This means that we do not require a special dataset for pre-training, and directly start our first stage by learning a subset of the target dataset.
>
> ### w.3: Memory costs
> We respectfully disagree. We do take into account the memory footprint taken by the global planes. However, since these planes are **shared**, their number is fixed and does not scale with the number of scenes. This means that we only have to take them into account once.
>
> ### q.1: 3D-aware latent space
>
> IG-AE is currently the only method proposing a latent space that is 3D-aware. This means that IG-AE preserves 3D-consistency between images when encoded into latent images.
> As such, it is currently the only available autoencoder with a latent space that is compatible with NeRF training.

---

> > ### Comment · Reviewer_8Gu4 · 2024-12-02
> >
> > I would to thank the authors for their extensive work during rebuttal on improving the quality of the evaluations and benchmarks. Specifically, I will call out the detailed comparison of training time, memory consumption, and rendering quality of their method versus standard scene-specific radiance fields (NeRF, I-NGP, etc) that were added to the supplementary.
> >
> > However, in its current state, I don't think this paper is ready for publication. Specifically, the experiments overall still need significant work to be comparable and glean useful insights.
> >
> > For the paper to be strengthened, I would re-iterate that there should be some of the following comparisons (which are still missing):
> > * A latent nerf comparison. I would disagree that there are no existing latent nerfs. I would consider [1] and [2] to be reasonable latent nerf baselines, or even feed-forward few-view methods such as PixelNeRF or PF-LRM and certainly could be run on the same object-centric setup that is currently evaluated.
> > * Fairer comparisons with per-scene optimization methods. I appreciate the authors evaluating standard NVS benchmarks (although 3DGS is still missing). The story is still very unclear.  I would say generally, the baselines have significantly better quality but longer training time and higher memory costs. Can you make these baselines more comparable by allocating the same amount of training time/memory for each method? For example, limit the training of each method to 5000 minutes total for all sequences, or limit the disk usage to 1GB (reducing the cache size, MLP size, number of Gaussian, etc accordingly). This would tell a clearer story of the trade-offs of the proposed method.
> >
> > I think these changes would significantly improve the paper and the evaluations.
> >
> > Re: memory consumption, thanks for the clarification. I think we are conflating active GPU memory vs disk usage, and the paper is making a claim that the total amount of disk needed to store all parameters for all sequences is significantly lower using the proposed approach. I appreciate this clarification, but am not convinced of its importance. Methods like 3DGS and Instant-NGP can give a reasonable reconstruction in realtime to a few seconds. If it only takes a few seconds to get a reasonable reconstruction, then there's no need to store all the past reconstructions on disk.
> >
> > [1] Tancik et al. Learned Initializations for Optimizing Coordinate-Based Neural Representations.
> >
> > [2] Hamdi et al. SPARF: Large-Scale Learning of 3D Sparse Radiance Fields from Few Input Images.

---

### Official Review · Reviewer_g87m · 2024-11-07

**Soundness:** 2
**Presentation:** 3
**Contribution:** 1
**Rating:** 1
**Confidence:** 4

**Summary:**

This paper introduces Scaled Inverse Graphics, a framework aimed at efficient large-scale 3D scene learning. The framework leverage a Micro-Macro decomposition and a two-stage training process, reducing memory use and training time. Using Tri-Planes in a latent-space auto-encoder, it achieves efficient scene representations by first learning shared features on a subset of scenes and then applying these to train remaining scenes. Experiments on ShapeNet Cars and Basel Faces validate the method’s resource efficiency and rendering quality.

**Strengths:**

1. The framework presents a unique decomposed scene representation, with Micro and Macro components, facilitating shared feature learning across a large dataset. This approach is novel in the context of NeRF-based methods, where scene-specific and globally shared information is seldom jointly optimized (in an explicit way).
2. The paper is well-written and well-organized. The methodology is detailed.

**Weaknesses:**

The major weakness of this paper is very limited experiments and analysis on the performance of the proposed method.
1. The experiments are limited to single-category datasets (e.g., cars and faces), where scene variations are relatively uniform. This narrow scope restricts its applicability for more diverse datasets with higher inter-sample variation. Without more experiments, it is impossible to fully assess the method. Demonstrating this method's capability to handle more complex, multi-category datasets would strengthen its generalizability claims.
2. The experiments are limited to object-level datasets, where the scene "scale" is relatively small.
3. The method is not competitive when compare to recent advances on feed-forward reconstruction pipelines (LRM-variants, multi-view 3D Gen), while the training and inference settings are similar. The latter works are more efficient and do not rely on slow volumetric rendering while handle sparse settings well.
4. The experiments lack relevant baselines (only compared to Tri-Planes), including recent advanced NeRF-variants, 3DGS-variants, Feed-forward nerf variants (MVSNeRF), and LRM-variants and 3D generative models with image condition.

**Questions:**

See weaknesses.

---

> ### Author Response · Authors · 2024-11-28
>
> We thank the reviewer for their constructive remarks.
>
> ### w.1 + w.2: Adopted scenes
> We refer the reviewer to our global answer, "Limitation to object-level, semantically similar scenes", [here](https://openreview.net/forum?id=GSckuQMzBG&noteId=L9CjDTGnsr).
>
> ### w.3: Comparison with feedforward methods
> We refer the reviewer to our global answer, "Comparison with feedforward baselines", [here](https://openreview.net/forum?id=GSckuQMzBG&noteId=L9CjDTGnsr).
>
> ### w.4: Comparisons with baselines
> We do compare with other NeRF variants, particularly, Instant-NGP, TensoRF, K-Planes, and Vanilla NeRF (figure 1, table 4).

---

### Author Response · Authors · 2024-11-28

We thank the reviewers for their constructive feedback which will be invaluable to improve our paper. We replied to each reviewer individually and address common concerns in this answer.

- **Comparison with feedforward baselines**: The reviewers have pointed out that feedforward methods could be compared with our method, and that they were not mentioned in our paper. We thank them for revealing this point. We will work on a comparison with such methods in a future version of our paper.
- **Limitation to object-level, semantically similar scenes**: Regarding this point, we would like to clarify that this is the target for the method. In fact, our method was designed from the ground up to handle such cases (e.g. learning a repertoire of shoes or hats for a retail application). This is our paper's motivation, and the inspiration behind leveraging the common structures (in the Macro planes) on a large dataset of semantically similar scenes.

---

### Public Comment · ~Karim_Kassab1 · 2025-02-06
**Updated version of this work**

We would like to share the updated version of our paper, now available on arXiv at the following link: **[Fused-Planes: Improving Planar Representations for Learning Large Sets of 3D Scenes](https://arxiv.org/abs/2410.23742v2)**. We thank the reviewers once again for their constructive feedback, which has been invaluable in improving our paper. Specifically, we have incorporated the following changes.

### Motivation and related work

- We have seperated our proposed method **"Fused-Planes"**, from our novel framework **"Multi-Scene Inverse Graphics" (MSIG)**.
- We have **revised the motivation** behind our work, emphasizing the importance and pertinence of MSIG in recent works. More particularly, the prevalent adoption of planar scene representations for MSIG motivates a new planar method such as FusedPlanes with improved resources costs.
- We have **expanded the related work section** to provide a more comprehensive discussion of both recent works directly relevant to our proposed method, and approaches that could be applied or have potential for MSIG (e.g. feed-forward methods). We have additionally provided a detailed overview of these works in Appendix A, used to select relevant baselines and datasets in our experiments (cf. below).

### Additional baselines and expanded experiments

- We have largely increased the number of our **experimental baselines**, relying on a **systematic selection based on our related work**. In particular, we compare Fused-Planes with all baselines with the same NVS setting as ours (many-views). Baselines now include:
    - **Single-scene baselines**: Vanilla-NeRF [1], INGP [2], TensoRF [3], 3DGS [4], and K-Planes [5];
    - **Many-scene baselines**: CodeNeRF [6], CodeNeRF (MSIG),$^1$ SPARF [7], and Tri-Planes [8].
- **Datasets.**
    - We have included **three additional datasets** from ShapeNet.
    - We have **justified our selection** of evaluation datasets: we utilize **the same datasets used by our direct baselines** (many-scene baselines, as identified in our related work), along with additional ones.

Overall, our method achieves state-of-the-art efficiency in MSIG resource costs among planar scene representations while maintaining similar rendering quality.

We remain open for any further discussions.

---

$^1$ CodeNeRF (MSIG) is a variant of CodeNeRF that we have adapted for multi-scene inverse graphics. More details are present in Appendix C.

### References

[1] Mildenhall, B., Srinivasan, P. P., Tancik, M., Barron, J. T., Ramamoorthi, R., & Ng, R. (2020). NeRF: Representing Scenes as Neural Radiance Fields for View Synthesis. ECCV.

[2] Müller, T., Evans, A., Schied, C., & Keller, A. (2022). Instant neural graphics primitives with a multiresolution hash encoding. ACM Transactions on Graphics (TOG), 41(4), 1–15.

[3] Chen, A., Xu, Z., Geiger, A., Yu, J., & Su, H. (2022). TensoRF: Tensorial Radiance Fields. European Conference on Computer Vision (ECCV).

[4] Kerbl, B., Kopanas, G., Leimkühler, T., & Drettakis, G. (2023). 3D Gaussian Splatting for Real-Time Radiance Field Rendering. ACM Transactions on Graphics, 42(4).

[5] Fridovich-Keil, S., Meanti, G., Warburg, F. R., Recht, B., & Kanazawa, A. (2023, June). K-Planes: Explicit Radiance Fields in Space, Time, and Appearance. Proceedings of the IEEE/CVF Conference on Computer Vision and Pattern Recognition (CVPR), 12479–12488.

[6] Jang, W., & Agapito, L. (2021, October). CodeNeRF: Disentangled Neural Radiance Fields for Object Categories. Proceedings of the IEEE/CVF International Conference on Computer Vision (ICCV), 12949–12958.

[7] Hamdi, A., Ghanem, B., & Nießsner, M. (2023, October). SPARF: Large-Scale Learning of 3D Sparse Radiance Fields from Few Input Images. Proceedings of the IEEE/CVF International Conference on Computer Vision (ICCV) Workshops, 2930–2940.

[8] Chan, E. R., Lin, C. Z., Chan, M. A., Nagano, K., Pan, B., De Mello, S., … Wetzstein, G. (2022, June). Efficient Geometry-Aware 3D Generative Adversarial Networks. Proceedings of the IEEE/CVF Conference on Computer Vision and Pattern Recognition (CVPR), 16123–16133.

---

### Public Comment · ~Karim_Kassab1 · 2026-03-12
**Updated version of this work (ICLR 2026)**

We would like to inform the readers and reviewers that an updated and extended version of this work has been accepted at ICLR 2026 under the title:

**[Fused-Planes: Why Train a Thousand Tri-Planes When You Can Share?](https://openreview.net/forum?id=bAG7lS1AUL)**

This new version expands upon the ideas presented in this submission and incorporates improvements motivated by the reviewers’ feedback, including stronger motivation, expanded comparisons with relevant baselines, and additional experiments.

More information about the updated work, including the accepted paper and open-source code, can be found on [our project page](https://fused-planes.github.io/).

We thank the reviewers once again for their constructive feedback, which helped improve the work.

---

### Meta-Review · Area_Chair_kLoQ · 2024-12-21

**Metareview:**

Reviewers maintain a unanimous recommendation for rejection, citing limited applicability of closed-domain experiments, weak motivation, and lack of comparison to current leading methods.

Nevertheless, reviewers appreciate the paper's novel ideas, and clear presentation. The authors are encouraged to read the reviewer's suggestions and resubmit a revised manuscript.

**Additional Comments On Reviewer Discussion:**

Reviews were unanimously negative. Reviewers mostly raised points about weak comparisons with state-of-the-art methods for feed-forward 3D prediction, e.g., LRM.

---

### Decision · Program_Chairs · 2025-01-22

Reject